# Correlation of C-arm CT acquired parenchymal blood volume (PBV) with 99mTc-macroaggregated albumin (MAA) SPECT/CT for radioembolization work-up

**Matthias Weissinger** [1], **Jonas Vogel** [1,2], **Jürgen Kupferschläger** [1], **Helmut Dittmann** [1], **Salvador Guillermo Castaneda Vega** [1,3], **Ulrich Grosse** [4], **Christoph Artzner** [2], **Konstantin Nikolaou** [2,5,6], **Christian la Fougere** [1,5,6☯] *, **Gerd Grözinger** [2☯]

1 Department of Nuclear Medicine and Clinical Molecular Imaging, University Hospital Tuebingen, Tuebingen, Germany, 2 Department of Diagnostic and Interventional Radiology, University Hospital Tuebingen, Tuebingen, Germany, 3 Department for Preclinical Imaging and Radiopharmacy, Werner Siemens Imaging Center, University Hospital Tuebingen, Tuebingen, Germany, 4 Department of Diagnostic and Interventional Radiology, Kantonsspital Frauenfeld, Frauenfeld, Switzerland, 5 iFIT-Cluster of Excellence, Eberhard Karls University Tuebingen, Tuebingen, Germany, 6 German Cancer Consortium (DKTK), Partner Site Tuebingen, Tuebingen, Germany

☯ These authors contributed equally to this work.
* christian.lafougere@med.uni-tuebingen.de

# Abstract

## Objective

SPECT/CT with 99mTc-macroaggregated albumin (MAA) is generally used for diagnostic work-up prior to transarterial radioembolization (TARE) to exclude shunts and to provide additional information for treatment stratification and dose calculation. C-arm CT is used for determination of lobular vascular supply and assessment of parenchymal blood volume (PBV). Aim of this study was to correlate MAA-uptake and PBV-maps in hepatocellular carcinoma (HCC) and hepatic metastases of the colorectal carcinoma (CRC).

## Materials and methods

34 patients underwent a PBV C-arm CT immediately followed by 99mTc-MAA injection and a SPECT/CT acquisition after 1 h uptake. MAA-uptake and PBV-maps were visually assessed and semi-quantitatively analyzed (MAA-tumor/liver-parenchyma = MAA-TBR or PBV in ml/100ml). In case of a poor match, tumors were additionally correlated with post-TARE 90Y-Bremsstrahlung-SPECT/CT as a reference.

## Results

102 HCC or CRC metastases were analyzed. HCC presented with significantly higher MAA-TBR (7.6 vs. 3.9, p<0.05) compared to CRC. Tumors showed strong intra- and inter-individual dissimilarities between TBR and PBV with a weak correlations for capsular HCCs (r = 0.45, p<0.05) and no correlation for CRC. The demarcation of lesions was slightly better for both HCC and CRC in PBV-maps compared to MAA-SPECT/CT (exact match: 52%/50%;

**Data Availability Statement:** All relevant data are within the manuscript and its Supporting Information files.

**Funding:** This study was funded in part by the Deutsche Forschungsgemeinschaft (DFG, German Research Foundation) under Germany's Excellence Strategy to CLF (EXC 2180 – 390900677) and the Open Access Publishing Fund of University of Tuebingen provided support for publishing fees. No additional external funding was received for this study. The funders had no role in study design, data collection and analysis, decision to publish, or preparation of the manuscript.

**Competing interests:** The authors have declared that no competing interests exist.

same intensity/homogeneity: 38%/39%; insufficient 10%/11%). MAA-SPECT/CT revealed a better visual correlation with post-therapeutic $^{90}$Y-Bremsstrahlung-SPECT/CT.

## Conclusion

The acquisition of PBV can improve the detectability of small intrahepatic tumors and correlates with the MAA-Uptake in HCC. The results indicate that $^{99m}$Tc-MAA-SPECT/CT remains to be the superior method for the prediction of post-therapeutic $^{90}$Y-particle distribution, especially in CRC. However, intra-procedural PBV acquisition has the potential to become an additional factor for TARE planning, in addition to improving the determination of segment and tumor blood supply, which has been demonstrated previously.

## Introduction

Transarterial radioembolization (TARE), also known as selective internal radiotherapy (SIRT) is a locoregional treatment option for patients suffering from unresectable primary and secondary tumors (metastases) of the liver [1, 2]. The therapy consists of $^{90}$Y-loaded resin or glass microspheres, which are infused via the hepatic artery to target the terminal arterioles of tumors. The primary mechanism of anti-tumoral action of TARE is local irradiation rather than stopping blood flow resulting from arterial embolization [3].

For the work-up of TARE a two-step approach is employed, starting with angiography and optional prophylactic coil-embolization of obvious shunt vessels. This is followed by injection of $^{99m}$Tc-labeled macroaggregated albumin particles ($^{99m}$Tc-MAA) in order to simulate the therapeutic microspheres distribution. SPECT/CT with $^{99m}$Tc-MAA is the method of choice to evaluate extrahepatic perfusion and assessing the hepato-pulmonary shunt fraction [4]. In addition, $^{99m}$Tc-MAA deposition is thought to be dependent on regional arterial vascularization and is therefore used to simulate microsphere distribution within the liver [5]. However, the role of $^{99m}$Tc-MAA deposition in the prediction of intra-therapeutic $^{90}$Y-microsphere-deposition [6], as well its correlation to absorbed tumor dose, tumor response and progression-free survival [7] is still discussed controversially.

Some studies have shown a moderate correlation between pre-therapeutic $^{99m}$Tc-MAA uptake and post-TARE $^{90}$Y-microsphere accumulation in the tumors [8], as well as progression free survival after TARE [7]. Moreover, $^{99m}$Tc-MAA SPECT/CT has been discussed to be an imperfect surrogate for $^{90}$Y-microsphere deposition [9, 10] due to its significant underestimation of the delivered radiation dose during TARE [8, 11].

Concordantly, Ilhan et al. [6] showed that approximately 60% of tumors demonstrating low $^{99m}$Tc-MAA uptake present a significantly higher uptake in the $^{90}$Y-Bremsstrahlung scans after TARE. This known underestimation of targeting might exclude patients from TARE or increase the risk of a radiation-induced liver disease (RILD), as a result of unnecessary dose escalation [12].

Due to the described limitation of $^{99m}$Tc-MAA-SPET/CT for predicting intrahepatic $^{90}$Y-spheres distribution [6], new approaches for a better prediction of tumor perfusion and thus $^{90}$Y-sphere deposition are needed.

A promising parameter for this purpose could be the assessment of parenchymal blood volume (PBV), which can be calculated using a dual phase C-arm CT protocol and offers the possibility to measure quantitatively gauge perfusion features of a specific liver tumor [13].

PBV has been shown to correlate with other parameters of perfusion measurement like volume perfusion CT [14] and to enable the visual and quantitative assessment of tumor-perfusion directly during the angiographic work-up procedure. Furthermore, the acquisition of 3D C-arm CTs improve the detection of aberrant vessels and identification of vascular territory supply compared to regular digital subtraction angiography images and has become part of the routine work-up in various centers [15–17]. Additionally, the benefit for the characterization of tumors and subsequent image-guided transarterial therapies has already been shown in several studies [16, 18, 19].

Therefore, the aim of this study was to evaluate if PBV-maps of liver parenchyma and tumors positively correlate with $^{99m}$Tc-MAA-SPECT/CT uptake and thus may be used as an additional tool to predict tumor targeting during the angiographic TARE procedure in order to deliver a more individualized intervention.

## Material and methods

This retrospective analysis was approved by the Ethics Committee of the University of Tuebingen (Decision No. 747/2014BO1). Informed consent for retrospective analysis of the data was acquired from all patient included into this study. Decision for TARE treatment was made by the interdisciplinary tumor board of our Comprehensive Cancer Center.

### Patient cohort

34 consecutive patients suffering from HCC (n = 19) and CRC liver metastases (n = 15) scheduled to receive a TARE were included between October 2014 and February 2016 in this retrospective study. All patients underwent C-arm CT and $^{99m}$Tc-MAA-SPET/CT examination prior to TARE. Detailed patient's characteristics are shown in Table 1.

Up to 5 representative tumors per patient were defined in the last contrast enhanced cross-sectional imaging and were evaluated in each patient on the basis of tumor size, best visibility and delineation (MRI n = 15, CT n = 18, $^{18}$F-FDG PET/CT n = 1, interval 38 ± 21 days). In total, 102 tumors (54 CRC, 48 HCC) were defined accordingly and analyzed further using C-arm CT and SPET/CT. Comprehensive tumor characteristics are shown in S1 Table in S1 Data. Mean age was balanced in both groups (65.4 ± 11.0 vs. 65.9 ± 8.2 years).

### Angiography

The angiography was performed by a team of three interventional radiologists, each with at least 5 years of experience in transarterial liver therapies. A robotic angiographic suite (Artis

**Table 1. Patients characteristics.**

|  | HCC | CRC | Total |
|---|---|---|---|
| n | 19 | 15 | 34 |
| Sex (f/m) | 1/18 | 4/11 | 5/29 |
| Age (years) | 65.9 ± 8.2 | 65.4 ± 11.0 | 65.7 ± 9.4 |
| Size (cm) | 173.2 ± 7.0 | 174.1± 10.6 | 173.6 ± 8.7 |
| Weight (kg) | 79.7 ± 12.1 | 76.9 ± 11.8 | 78.4 ± 11.9 |
| BMI | 26.7 ± 4.4 | 25.04± 3.8 | 26.1 ± 4.1 |
| $^{99m}$Tc-MAA activity whole liver (MBq) | 136.8 ± 32.6 | 127.5 ± 44.2 | 132.7 ± 37.8 |
| $^{99m}$Tc-MAA activity per hepatic lobe | 76.4 ± 28.0 | 70.6 ± 23.6 | 74.4 ± 27.1 |
| Number of measured lesions | 48 | 54 | 102 |
| Lesion size (long axis in mm) | 38.3 ± 32.1 | 33.8 ± 22.7 | 35.9 ± 27.4 |

Zeego Q®, VE40A, Siemens Healthineers, Forchheim, Germany) was used for all planning angiograms and following TARE procedures, Selective right and left hepatic angiography was performed using a 2.7-French microcatheter system (Progreat®, Terumo, Leuven, Belgium) to evaluate variants of hepatic anatomy and subsequent prophylactic embolization of extrahepatic vessels.

A separate angiogram was performed via the microcatheter for each simulated catheter position.

## C-arm CT and post-processing

C-arm CT was performed routinely during the TARE work-up intervention on the same angiography suite after the planning angiogram for each simulated catheter position prior to $^{99m}$Tc-MAA-injection. C-arm CT consisted of unenhanced rotation (mask run) and contrast enhanced rotation (fill run) for the acquisition of parenchymal blood volume (PBV) maps (time per rotation 4s, total examination time 16s, 90 kV, 200° total angle, 0.8° per frame, 248 frames, matrix 616x480 pixel, flat panel size 616μm, dose 0.36μGy per frame) [18, 19]. Contrast agent Iopromide (7.5ml Ultravist 370 (Bayer Schering, Leverkusen, Germany) diluted with 22.5ml NaCL 0.9%) was injected by an automated power injector (Accutron-HP-D, Medtron, Saarbrücken, Germany), using a flow rate of 2ml/s. Contrast injection was performed immediately after the mask run and was started manually to acquire a contrast enhanced acquisition in a steady-state of liver perfusion according to previous studies investigating PBV of the liver [13, 20]. All acquired data were post-processed and motion corrected on a commercially available workstation (Syngo XWP, Siemens Healthineers) using an automatic reconstruction algorithm as previously described [21].

## $^{99m}$Tc-MAA-SPECT/CT

Perchlorate (600mg) was administered prior to $^{99m}$Tc-MAA-injection (TechneScan®Lyo-MAA, Mallinckrodt Pharmaceuticals, Surrey, GB).$^{99m}$Tc-MAA (avg. 132.7 ± 37.8 MBq, for further details see Table 1) was injected as 1ml single bolus over 5 seconds into the arterial branches supplying the liver area to be treated, and flushed by 5ml saline (one-sided MAA-application in 6/34 patients). SPECT/CT was performed with a dual-headed SPECT/CT gamma camera (GE Discovery 670 Pro®; GE Healthcare, Chicago Il, USA) within one hour from $^{99m}$Tc-MAA-injection (SPECT/CT scan parameters are shown in Table 2). CT was performed for anatomical mapping and attenuation correction.

**Table 2. SPECT/CT scan parameters.**

|  | $^{99m}$Tc-MAA scan | $^{90}$Y-Bremsstrahlung scan |
|---|---|---|
| **SPECT** |  |  |
| Collimator | low energy | high energy |
| Field-of-view (bed positions) | 76cm (2) | 40cm (1) |
| Covered body region | thorax + abdomen | abdomen |
| Matrix | 128x128 | 128x128 |
| SPECT steps | 30 | 30 |
| Acquisition time per 6° step | 20sec | 20sec |
| **CT** |  |  |
| Tube Current (mAs) | 80–220 mAs | 10–80 mAs |
| Tube voltage | 120 KV | 120KV |
| Slice thickness | 2.5mm | 2.5mm |

SPECT-images were reconstructed with an OSEM iterative reconstruction protocol (2 iterations, 10 subsets). Quantitative SPECT/CT data were post processed with a dedicated software algorithm (Evolution®, GE Healthcare, Chicago, USA) and co-registered with CT images (GE Xeleris 3®, GE Healthcare, Chicago, USA).

## Image evaluation

Up to five representative intrahepatic tumors per patient were defined as described above. Tumors located alongside coils were avoided due to possible beam hardening artifacts. Perpendicular diameters were measured in cross-sectional images for all modalities accordingly. For quantitative analysis, regions-of-interest (ROIs) for PBV and volumes-of-interest (VOIs) for $^{99m}$Tc-MAA-SPEC/CT were defined manually and separately for each modality according to the respective functional images. Because of unavoidable variations in the ratio of injected $^{99m}$Tc-MAA to the mass of the treated liver segment, the ratio of $^{99m}$Tc-MAA-uptake of the tumor to healthy liver tissue background (hereafter abbreviated as TBR), instead of absolute values, was assessed for further evaluations.

Moreover, in order to ensure a high conformity to the tumor-morphology, every VOI was coregistered to the corresponding pre-interventional CT or MRI and carefully adapted in size if necessary. Background ROIs ($>3cm^2$ for PBV) and VOIs ($>50ml$ for SPECT/CT) were placed in non-tumorous liver tissue within each individual vascular territory of the corresponding catheter position.

The tumors were classified into two subgroups based on their tumor margins and parenchyma infiltration in the pre-therapeutic cross sectional images. More specifically, tumors with diffuse (diffuse margins, diffuse growth into liver parenchyma) and capsular (focal nodular, clear demarcation, encapsulated, less infiltrative) growth pattern were defined according to the criteria previously described in the literature [22, 23].

Additionally, each tumor was classified visually according to its intensity and perfusion homogeneity using a 6 point optical lesion characteristics scale (OLC) which is defined and visualized in the Fig 1A and 1B.

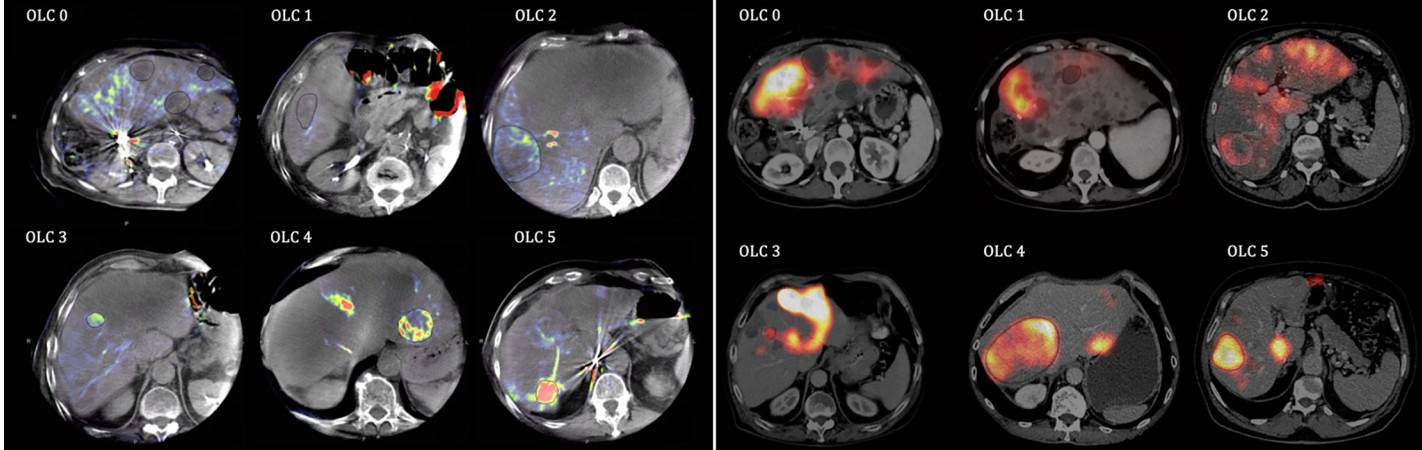

**Fig 1.** 6 point OLC scale for PBV-map (a) and 6 point OLC-Scale for $^{99m}$Tc-MAA-SPECT/CT (b). Tumors were classified according to their heterogeneity and intensity using a visual 6 point scale (OLC) for PBV /$^{99m}$Tc-MAA uptake compared to healthy liver tissue. OLC 0: PBV/uptake < normal liver tissue. OLC 1: PBV/uptake = normal liver tissue. OLC 2: PBV/uptake slightly increased inhomogeneous perfusion/distribution. OLC 3: PBV/uptake slightly increased homogeneous perfusion/distribution. OLC 4: PBV/uptake clearly increased inhomogeneous perfusion/distribution. OLC 5: PBV/uptake clearly increased homogeneous perfusion/distribution.

## $^{90}$Y-Bremsstrahlung-SPECT/CT

Tumors with an OLC mismatch between $^{99m}$Tc-MAA-SPEC/CT and the PBV-map were correlated retrospectively with the post-therapeutic $^{90}$Y-Bremsstrahlung-SPECT/CT as a reference.

The $^{90}$Y microspheres were applied in each simulated catheter position separately (1238 ± 603 MBq $^{90}$Y per lobe). The angiographic catheter was placed very carefully in exactly the same position as during the $^{99m}$Tc-MAA injection.

SPECT/CT acquisition was performed within 24 hours after TARE with the SPECT/CT scan parameters as listed in Table 2. $^{90}$Y-Bremsstrahlung scans were reconstructed and post-processed with the same software as used for $^{99m}$Tc-MAA-SPECT/CT (described above). Due to the limited resolution of the $^{90}$Y-Bremsstrahlung scan, only tumors ≥ 25 mm were evaluated.

### Statistics

Tumor size, TBR and MAA-Uptake was log-normal distributed thus the statistical tests were performed on natural log-transformed data (histograms presented as S1a-S1d Fig in S1 Data). For the comparison of PBV and $^{99m}$Tc-MAA-Upake or TBR, linear regression with robust clustered standard error correction was applied using Stata 14 (StataCorp LLC, College Station, Texas, USA). Bland-Altman plots were calculated to analyze the agreement of tumor size measurements using SPSS (Version 27, IBM Corporation, Armonk, New York, USA). Exploratory data analysis was performed using SPSS and JMP® (Version 13.1, SAS Institute Corporation, Heidelberg, Germany).

Because of the data structure, for the determination of the overall within-individual relationship among paired measures (PBV and TBR) within one patient, repeated measures correlations were performed with multiple measurement correction using rmcorr-package for R (V0.4.1.by J.Bakdash and L. Marusich)

Rules of thumb was applied to interpret correlation coefficient rho: 0.20–0.39: weak; 0.40–0.59: moderate; 0.60–0.79: strong; 0.80–1.0: very strong.

Significance level of P values was 0.05. All values are expressed as mean values ± standard deviations with 95% confidence intervals are given in brackets.

## Results

### Tumor size

PBV-maps enabled an accurate assessment of the tumor size, with a good agreement to the pre-therapeutic CT and MRT-scans (C-arm CT 36.7 ± 27.2mm vs. CT/MRT 35.9 ± 24.4mm). The average difference of all tumor lesions was 0.8mm ± 7.7mm. Time interval between both scans was identified as the most important factor of influence (period 21 ± 16 days: 0.1mm ± 8.8mm; period 53 ± 11 days: 1.7mm ± 6.3mm).

In particular, CRC are displayed slightly too small in PBV maps compared to CT or MRI with an average discrepancy of 1.5mm as presented in Fig 2A. However, differences of measurements are within relative narrow limits (-7.0 to 10.0mm) and without a trend as mean of both measurements increase.

Measuring tumor size of HCC seems to have a perfect agreement with a mean difference of 0.1mm between both methods (Fig 2B). However, due to single outliers, the range for the 1.95 standard deviation limit was quite large (from -19.8 to 20.1mm), and was even exceeded by some outliers in both directions. However, most of the differences are within a narrow limit and without a trend increasing size related measurement differences or variabilities. Although our data are not distributed normally, the differences seem to be.

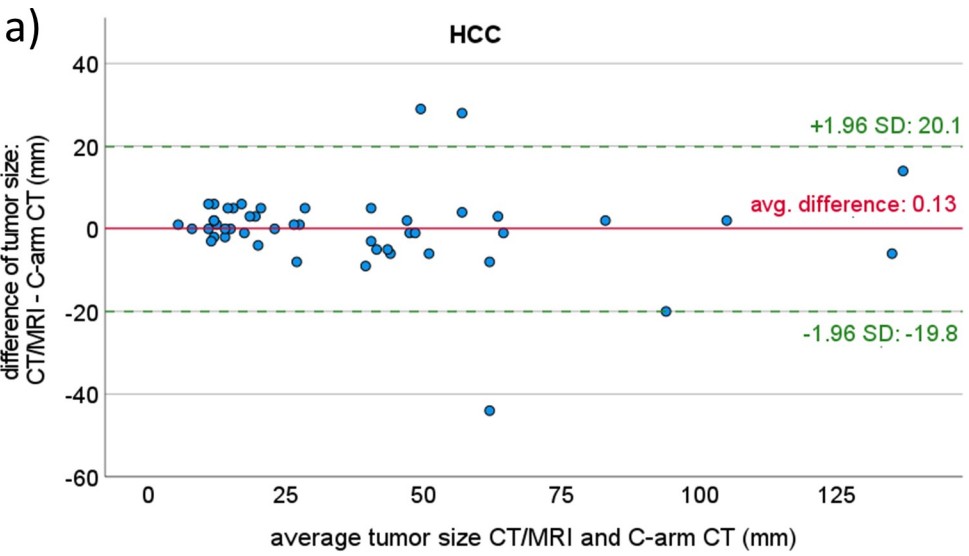

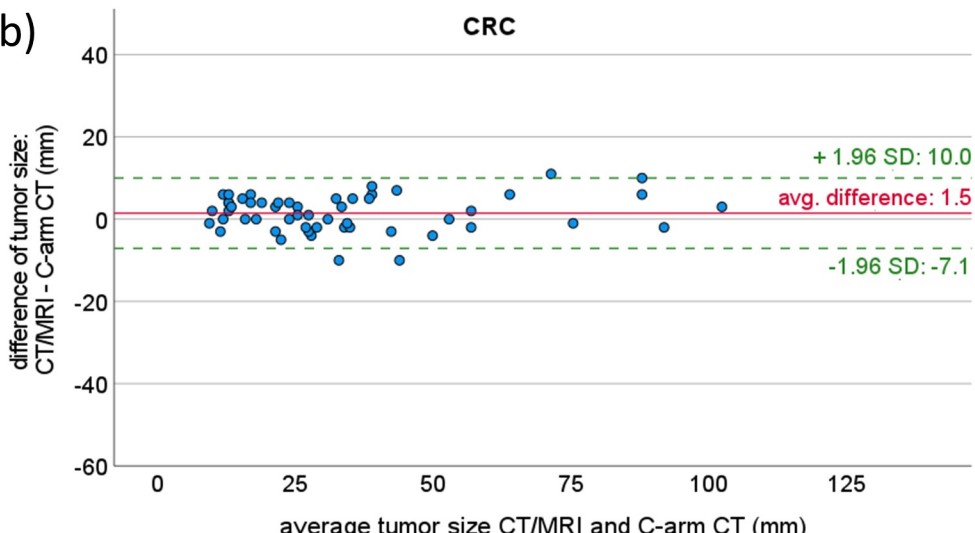

**Fig 2.** Agreement of tumor size measurement with PBV C-arm CT and pre-therapeutic conventional imaging with CT or MR visualized with Bland-Altman plot for CRC (a) and HCC (b).

Measuring the exact tumor size according to the $^{99m}$Tc-MAA-scan was biased by a priori non-definable radiotracer uptake thresholding that in fact was orientated on available morphological imaging data.

## Optical lesion characterization of PBV and $^{99m}$Tc-MAA-scans

Overall, the tumors exhibited similar OLC values for both PBV-maps and $^{99m}$Tc-MAA-scans (Table 3). For both methods, HCC as well as CRC tumors presented with a predominantly high PBV/uptake with either homogeneous or inhomogeneous perfusion (OLC 4–5: 60% and 46% correspondingly).

**Table 3. Optical lesion characteristic score (OLC) for PBV C-arm CT and $^{99m}$Tc-MAA-SPECT/CT.**

| | | OLC (PBV-scan) | | | | | | | OLC ($^{99m}$Tc-MAA-SPECT/CT) | | | | | | |
|---|---|---|---|---|---|---|---|---|---|---|---|---|---|---|---|
| | | 0 | 1 | 2 | 3 | 4 | 5 | total | 0 | 1 | 2 | 3 | 4 | 5 | total |
| **Tumor entity** | | | | | | | | | | | | | | | |
| | HCC | 0 | 1 | 3 | 15 | 9 | 20 | 48 | 0 | 3 | 4 | 12 | 6 | 23 | 48 |
| | CRC | 6 | 2 | 7 | 14 | 15 | 10 | 54 | 4 | 5 | 1 | 18 | 6 | 20 | 54 |
| **Growth pattern** | | | | | | | | | | | | | | | |
| | capsular | 4 | 1 | 5 | 16 | 13 | 15 | 54 | 2 | 4 | 5 | 16 | 7 | 20 | 54 |
| | diffuse | 2 | 2 | 5 | 13 | 11 | 15 | 48 | 2 | 4 | 0 | 14 | 5 | 23 | 48 |
| **Tumor size** | | | | | | | | | | | | | | | |
| | ≤ 25 mm | 3 | 1 | 1 | 18 | 8 | 15 | 46 | 1 | 6 | 0 | 19 | 0 | 20 | 46 |
| | > 25 mm | 3 | 2 | 9 | 11 | 16 | 15 | 56 | 3 | 2 | 5 | 11 | 12 | 23 | 56 |
| **Total** | | 6 | 3 | 10 | 29 | 24 | 30 | 102 | 4 | 8 | 5 | 30 | 12 | 43 | 102 |

Visual grading of PBV-maps and $^{99m}$Tc-MAA scans revealed a higher rate of clearly increased perfusion/uptake in HCC compared to CRC. Despite the good overall consistency in OLC evaluation between both methods, the rate of homogeneous intense perfusion/uptake (OLC 5) was lower in PBV compared to $^{99m}$Tc-MAA.

More specifically, in PBV-maps, HCCs presented with a higher rate of clearly increased tumor perfusion compared to CRCs (OLC 4–5: 60% vs. 46%) and a lower rate of non-increased tumor uptake (OLC 0–1: 2% vs. 15%).

Similar results were found for $^{99m}$Tc-MAA-scans with a visually higher uptake of HCCs than CRCs (OLC 4–5: 60% vs. 48%; OLC 0–1: 6% vs. 17%) as shown in detail in Table 3.

OLC evaluation of PBV and $^{99m}$Tc-MAA values were consistent (exact: 52/102) same category of intensity or homogeneity: 37/102), independent of tumor size, growth pattern or tumor entity. However, the amount of homogeneous intense uptake within the metastases (OLC 5) was less pronounced for the PBV-maps (30/102) when compared to $^{99m}$Tc-MAA-scans (43/102). A representative example for divergent OLCs and the advantages of a separate tumor mapping for each catheter position are shown in Fig 3.

A total of 23 tumors (11 HCC, 12 CRC) with a mismatch between PBV und $^{99m}$Tc-MAA-uptake were additionally correlated with the $^{90}$Y-Bremsstrahlung-SPECT/CT post TARE using the visual OLC-scale. TARE was performed in 13 patients with resin microspheres (Sirtex Medical, Sydney, Australia) and in one patient with glass microspheres (MDS Nordion, Kanata, Canada).

In case of a mismatch between the two simulation methods, $^{99m}$Tc-MAA-SPECT/CT revealed a markedly better prediction of the post-therapeutic $^{90}$Y-sphere distribution than the PBV-maps. This effect seems to be stronger in HCCs than in CRCs as shown in Table 4. A representative example is shown in Fig 4.

## Semi-quantitative PBV and $^{99m}$Tc-MAA-uptake assessment

The applied activity for $^{99m}$Tc MAA-SPECT CT was comparable in patients with HCC and CRC (see Table 1). Significance was tested on the log-transformed data, which are shown in S2 Table in S1 Data. Mean $^{99m}$Tc-MAA-TBR values were shown to be significantly higher in HCC compared to CRC as shown in Table 5. Furthermore, HCC presented with a higher intertumoral variability in both measures, thus indicating broad differences in tumor vascularization. The capsular HCCs had two outliers with 846 and 719kBq/cm$^3$ (S2 Fig in S1 Data). Smaller tumors (<25mm) presented only a non-significant trend towards higher PBV-values (p = 0.12). A diffuse growth pattern of the tumor was associated with a significantly higher

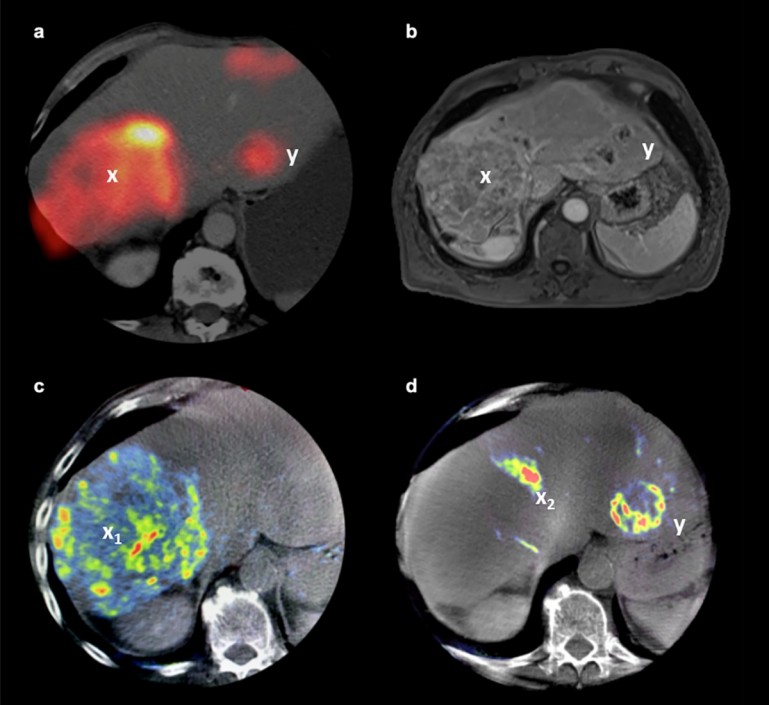

**Fig 3. Identification of tumor vascularization using PBV-map.** Example of a 63 year old patient suffering from multifocal HCC. $^{99m}$Tc-MAA-injection and C-arm CT was performed separately for each lobe via individual catheter positions. The images show the $^{99m}$Tc-MAA-SPECT/CT (a), contrast-enhanced T1w MRI (b) and PBV-maps (c, d). The SPECT/CT scan on panel "a" shows the outcome of the combined MAA-applications via both catheters. The PBV-maps on the other hand enable a mapping of the blood supply of the tumors (c, d) for each catheter position separately (markers $x_1$ and $x_2$). While SPECT/CT and MRI demonstrate a heterogeneous tumor bulk in Segment VIII (a, b), the PBV-map further details a separation of the HCC among its blood supply in a heterogeneously vascularized tumor region ($x_1$) in segment VIII as shown in panel "c" and a highly vascularized part ($x_2$) related to segment IV which is fed via the left catheter position (d). Another tumor (y) fed via the left catheter in Segment II (d) presents a clearly increased and homogeneous MAA-uptake (OLC 5) in SPECT/CT (a) and a clearly increased but heterogeneous perfusion (OLC 4) in the PBV-map (d).

$^{99m}$Tc-MAA background (normal liver tissue) when compared to a capsular growth pattern, resulting in lower $^{99m}$Tc-MAA-TBR values.

PBV values showed a weak but significant correlation to $^{99m}$Tc-MAA-TBR in HCC with capsular growth pattern (r = 0.45, p<0.05), but no correlation in CRC-metastases independent of the growth pattern (r = 0.1, p = 0.54) as presented in Fig 5. HCC with diffuse growth pattern presented with negative correlation of tumors within the same patient but weak overall correlation (r cluster corrected: -0.18, r overall: 0.21)

**Table 4. Visual correlation of $^{90}$Y-Bremsstrahlung-SPECT/CT with $^{99m}$Tc-MAA-SPEC/CT and PBV C-arm CT for tumors with lesion size $\geq$ 25 mm and mismatch in OLC between $^{99m}$Tc-MAA-SPEC/CT and PBV-C-arm CT.**

|  | Total (n = 23) | HCC (n = 11) | CRC (n = 12) |
|---|---|---|---|
| Better correlation with $^{99m}$Tc-MAA-SPEC/CT | 52% | 73% | 33% |
| Better correlation with PBV-C-arm CT | 17% | 18% | 17% |
| Equal correlation with $^{99m}$Tc-MAA-SPEC/CT and PBV-C-arm CT | 9% | - | 17% |
| Image quality of $^{90}$Y-Bremsstrahlung-SPECT/CT insufficient, no correlation possible | 22% | 9% | 33% |

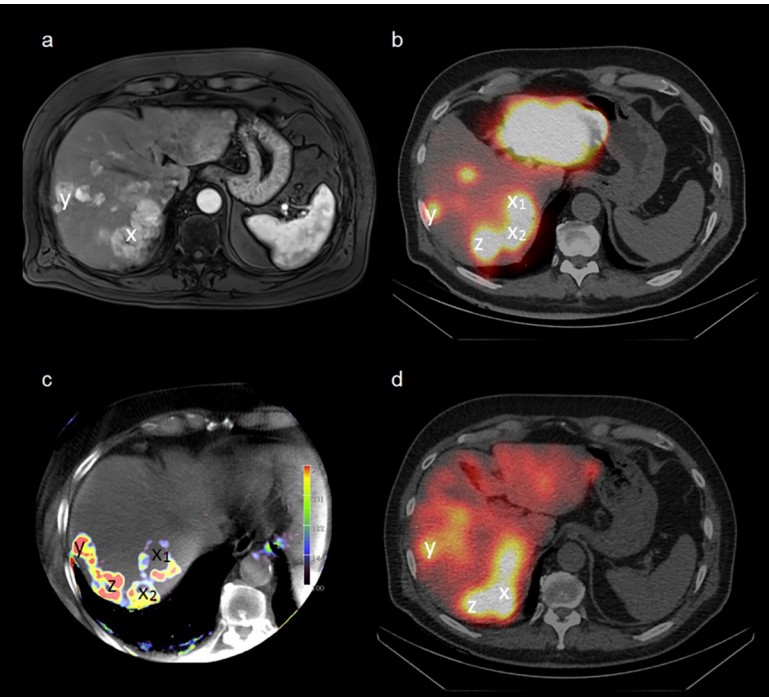

**Fig 4. Agreement of pre-therapeutic imaging with post-therapeutic $^{90}$Y-Bremsstrahlung-SPECT/CT.** Example of a 71-year-old patient suffering from multilocular HCC. The images show the contrast-enhanced T1 vibe MRI (a), $^{99m}$Tc-MAA-SPECT/CT (b), PBV-map (c) and post-therapeutic $^{90}$Y-Bremsstrahlung-SPECT/CT (d). Good visual agreement of $^{99m}$Tc-MAA uptake and PBV regarding the small lesion in segment VIII (y), but not for the bigger lesion in segment VII (x), especially the ventral part of the lesion ($x_1$). The third lesion (z) is not displayed in MRI due to smaller slice thickness and the more cranial position.

**Table 5. PBV and $^{99m}$Tc-MAA-uptake as measured in subgroups of tumor entity, tumor size and growth pattern.**

|  | C-arm CT | $^{99m}$Tc-MAA-SPECT/CT | | | n |
|---|---|---|---|---|---|
|  | PBV$_{lesion}$ [ml/100 ml] | MAA-TBR | U$_{lesion}$ [kBq/cm$^3$] | U$_{background}$ [kBq/cm$^3$] |  |
| **Average** | 11.3 (9.7–12.8) | 5.7 (4.6–6.7) | 76.0 (54.2–97.8) | 15.3 (13.0–17.6) | 102 |
| **Tumor entity** |  |  |  |  |  |
| HCC | 14.0 (11.4–16.7) | **7.6 $^+$ (5.6–9.6)** | 99.6 (54.7–144.5) | 15.5 (11.2–19.8) | 48 |
| ■ capsular | 13.3 (10.2–16.5) | 7.8 (5.3–10.4) | 101.5 (32.2–170.8) | 13.3 (8.1–18.6) | 31 |
| ■ diffuse | 15.3 (10.1–20.5) | 7.2 (3.7–10.6) | 96.3 (66.9–125.6) | 19.6 (11.8–27.4) | 17 |
| CRC | 8.8 (7.3–10.3) | **3.9 $^+$ (3.4–4.5)** | 55.0 (45.2–64.8) | 15.1 (12.8–17.3) | 54 |
| ■ capsular | 8.1 (5.8–10.5) | 4.2 (3.4–5.0) | 45.3 (31.9–58.6) | 11.7 (8.2–15.2) | 23 |
| ■ diffuse | 9.3 (7.3–11.3) | 3.8 (2.9–4.6) | 62.2 (48.3–76.2) | 17.6 (14.9–20.4) | 31 |
| **Growth pattern** |  |  |  |  |  |
| ■ capsular | 11.1 (9.0–13.3) | 6.3 (4.7–7.8) | 77.6 (37.6–117.5) | **12.6˚ (9.3–15.9)** | 54 |
| ■ diffuse | 11.4 (9.1–13.7) | 4.9 (3.6–6.3) | 74.2 (60.4–88.2) | **18.3˚ (15.2–21.5)** | 48 |
| **Tumor size** |  |  |  |  |  |
| ≤ 25 mm | 13.0 (10.4–15.5) | 6.0 (4.2–7.8) | 91.6 (45.3–137.9) | 16.9 (12.8–21.1) | 46 |
| > 25 mm | 9.9 (8.0–11.7) | 5.4 (4.2–6.6) | 63.2 (50.3–76.2) | 13.9 (11.4–16.4) | 56 |

Mean values with 95%CI in brackets are given. Logarithmic transformed data are presented in S2 Table in S1 Data.

Significant differences of log-transformed data are marked bold. Significant effects were found for ln(MAA-TBR)$^{(+)}$ between HCC and CRC (p$^{(+)}$ = 0.03) and for ln (U$_{background}$) in the liver in case of capsular or diffuse tumor growth$^{(˚)}$ p$^{(˚)}$ = 0.022.

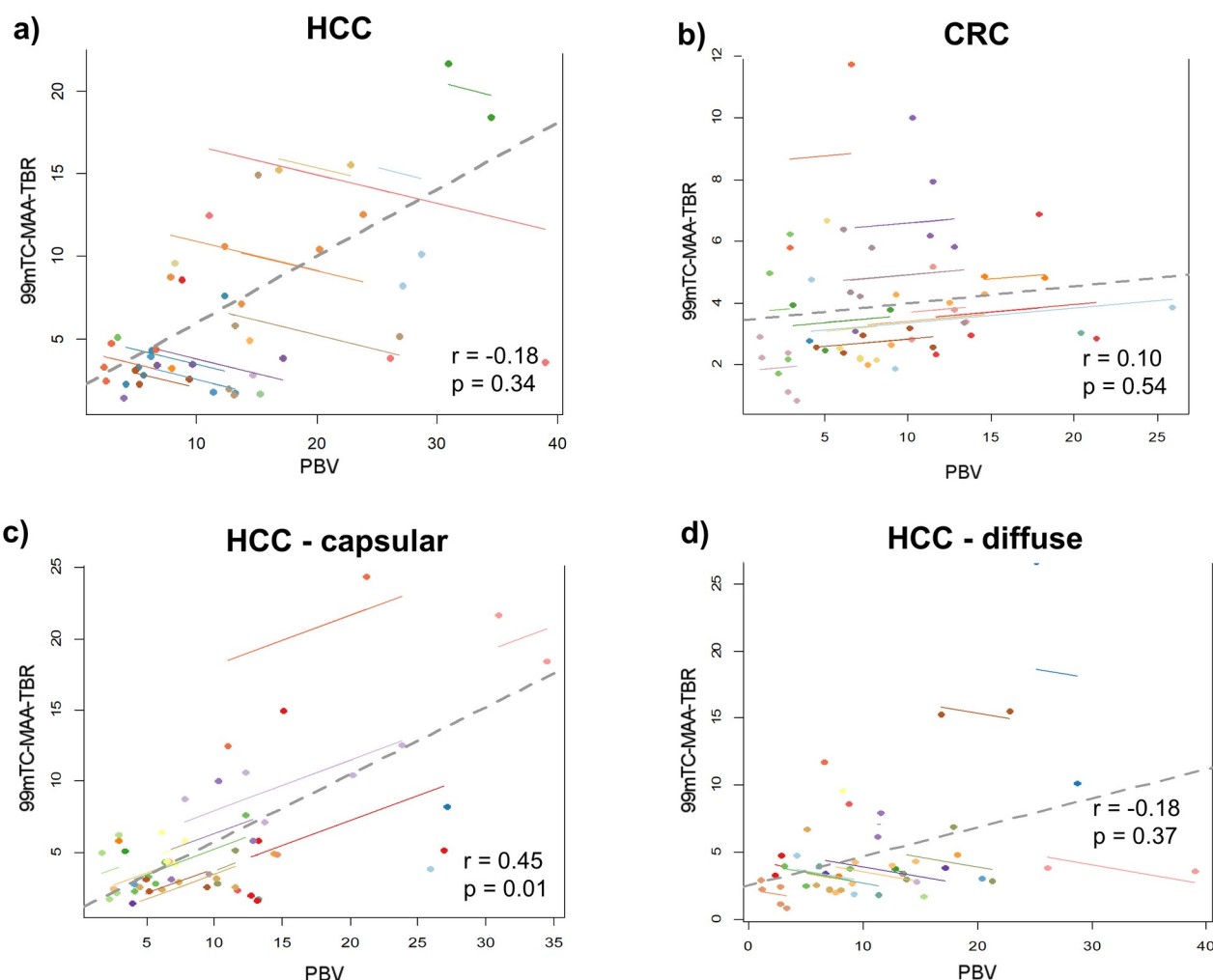

**Fig 5. Correlation between PBV and $^{99m}$Tc-MAA-TBR in different tumor entities and their growth pattern.** Moderate correlation between parenchymal blood volume (PBV) and $^{99m}$TC-MAA-TBR SPECT/CT was found for HCC (a), but not for CRC (b). Growth pattern of HCC influenced the correlation between both methods with a strong correlation in the capsular (c) and only a trend for a positive correlation for diffusely growing HCC (d).

## Discussion

This is the first study directly comparing C-arm CT based PBV measurements and $^{99m}$Tc-MAA-SPECT/CT distribution in patients suffering from HCC and CRC in a pre-therapeutic TARE setting. This study confirms the presumed higher perfusion of HCCs with significantly higher $^{99m}$Tc-MAA-TBR compared to CRC [6, 10].

In fact, we measured generally higher $^{99m}$Tc-MAA-TBR values compared to previous findings by Ilhan et al. [6] (HCC: 7.6 ± 6.8 vs. 2.11 ± 1.25; CRC: 3.9 ± 2.1 vs.1.80 ± 0.92). Furthermore, our intratumoral PBV measurements were also generally in-line with previously published data [18, 24] (Table 6). The differences between the magnitudes of PBV between studies might be explained by slightly different robotic angiographic suite and image post-processing approaches (e.g. software, ROI-definition). However, here, we report data consistently acquired with the same clinically approved software in a standardized protocol as previously described [18].

**Table 6. PBV values (in ml/100ml) as reported in previous studies.**

|  | PBV$_{all}$ | PBV$_{HCC}$ | PBV$_{CRC}$ |
|---|---|---|---|
| Syha et al. | - | 18.3 ± 6.2 | No CRC |
| Vogl et al. | 7.5 ± 5.6 | 9.9 ± 9.2 | 6.4 ± 4.0 |
| Weissinger et al. | 11.3 ± 7.9 | 14.0 ± 9.2 | 8.8 ± 5.6 |

The OLC analyses indicate that HCCs are associated with a clearly higher uptake both in PBV and MAA in comparison to CRC metastases.

These findings are consistent with CT perfusion studies showing higher arterial perfusion of HCCs compared to CRC metastases [25, 26].

Our study revealed significant differences between the distribution patterns of the contrast agent Iopromide used for PBV and the $^{99m}$Tc-MAA particles for SPECT/CT.

This was observed not only in tumors of various patients, but also with individual tumors of the same patient. Only HCCs with capsular growth patterns showed a weak correlation between the measurement methods.

Since C-arm CT and $^{99m}$Tc-MAA injection were performed almost simultaneously and without catheter dislocation, potential biases caused by incongruent positions of the intra-arterial catheter were negligible in our setup. This can be considered a major advantage of our approach, as such problems have been reported in previous studies without simultaneous PBV-imaging [6, 27, 28]. In addition, the acquired CT data ensure an accurate transfer of the tumor segmentation between PBV-maps and $^{99m}$Tc-MAA-scans.

One appropriate explanation might be the known gross difference in molecular weight between the contrast agent Iopromide and macro-aggregated albumin [29, 30]. The difference in molecular weight is so massive that perfusion, distribution and deposition, disregarding affinity, should be different.

It may be that Iopromide allows a perfusion measurement and very subtle permeability changes due to its fast kinetics, while $^{99m}$Tc-MAA represents only the perfusion of the larger and more organized capillary vessels and small arterioles (<20μm) [30].

Since $^{99m}$Tc-MAA may be restricted to a certain vessel size, a mismatch with Iopromide could be an indication of differences in the vascular structure of tumors and could provide further information on individual tumor perfusion.

This may also explain the poor correlation between PBV and MAA-uptake in CRC metastases. In these tumors, MAA-uptake is predominantly homogeneous, while PBV-maps demonstrate a significantly stronger inhomogeneity. This could indicate a stronger heterogeneity of small and disorganized tumor capillaries in CRC compared to HCC, which cannot be imaged by the large MAA particles. This thesis is supported by the results of Kim et al. [31], who found a significant difference in microvessel density in CRC depending on tumor differentiation. However, a higher microvessel density led to a significant decrease in blood flow in CT perfusion, indicating disorganized vessel structures [31]. Histological evidence of microvessel patency and organization, in correlation to PBV-maps and MAA-uptake, would be required to further test this hypothesis.

Furthermore, the comparably lower perfusion and high inter-individual variability of CRC metastases might cause high relative errors despite low differences in the absolute perfusion measurement [32].

The visual correlation between both methods also reveals that the size of the tumor may have an important impact. Although there was a slightly lower amount of tumors with intense and homogeneous contrast-agent distribution in the PBV-maps (29%) compared to $^{99m}$Tc-

MAA-SPETC/CT (42%), a higher amount of intense and inhomogeneous tumors (24% vs. 12%) was found in tumors ≤25mm. This can be explained by the lower resolution of SPECT/CT compared to C-arm CT (128x128 vs. 616x480 matrix). As a consequence, small tumors <25mm seem to be more homogeneous [33, 34]. As tumor heterogeneity is an important prognostic factor and associated with the progression-free survival [35–37], additional evaluation with PBV may help predict tumor targeting more precisely.

Furthermore, the partial volume effect [38] plays a strong role in small tumors, since SPECT/CT can cause an underestimation of the potential TARE-dose [6, 12]. As a consequence, patients might be excluded from a potentially life prolonging TARE.

As $^{99m}$Tc-MAA is generally considered to be an imperfect surrogate for the TARE dosimetry [9], we additionally analyzed the prediction of MAA particles and PBV-maps on post-therapeutic $^{90}$Y-microsphere distribution. Here $^{99m}$Tc-MAA-SPEC/CT imaging showed an overall better prediction of the post-therapeutic $^{90}$Y-sphere distribution than the PBV C-arm CT in the visual OLC analysis. This might be related to the closer similarity of $^{90}$Y-microsphere and MAA particles regarding size, weight and number of particles compared to Iopromide. However, the statistical power was limited due to the small sample size. Moreover, a direct quantitative correlation between PBV and post-therapeutic $^{90}$Y-spheres depositions (and thus the assessment of a potential advantage of PBV) could not be implemented in this study because of the small number of large tumors needed for a reliable quantification via $^{90}$Y-Bremsstrahlung-SPECT/CT.

Furthermore, the impossibility of an absolute quantification of the $^{99m}$Tc-MAA-uptake must be mentioned as a limitation. Although $^{99m}$Tc-MAA tumor-to-background ratio was calculated analogous to previous studies [6, 28] a quantitative assessment of the radiotracer uptake (in Bq/ml or SUV) would have been preferable. However, accurate data quantification would have been extremely complex and was hindered in the clinical setting due to uncertainties in the extent of the vascular territory in which the radiotracer was injected; especially in patients with multiple catheter positions. As a further minor limitation, it should be mentioned that the tumor quantification in the PBV maps could only be measured as a two dimensional region-of-interest due to software limitations. This may have led to slightly higher variances in quantification measurement.

## Conclusion

Parenchymal blood volume maps acquired by C-arm CT can improve the identification of small intrahepatic tumors and is only comparable to $^{99m}$Tc-MAA in patients with capsular HCCs. The additional assessment of tumor PBV during TARE-work-up allows a direct detection of tumor targeting during the planning procedure, especially in regard to a specific catheter position.

The post-therapeutic $^{90}$Y distribution in CRC metastases is more accurately predicted by $^{99m}$Tc-MAA-SPECT/CT than PBV-maps. Therefore our data supports that $^{99m}$Tc-MAA-SPECT/CT is the method of choice for personalized TARE planning. Further evaluations of quantitative $^{90}$Y post-TARE scans with improved resolution, tumor response and ultimately patient survival are further warranted.

Nonetheless, our data also suggests that PBV provides non-redundant perfusion information, which we hypothesize is dependent on micro-vessel architecture.

## Supporting information

**S1 Data.**
(DOCX)

## Acknowledgments

For the revision of this paper the methodological advice of the Institute of Clinical Epidemiology and Applied Biometry of the University of Tuebingen was consulted. The authors would like to thank Ms. You-Shan Feng for her capable support.

We acknowledge the Deutsche Forschungsgemeinschaft (DFG, German Research Foundation) under Germany's Excellence Strategy—EXC 2180–390900677 for supporting our study. We acknowledge support by Open Access Publishing Fund of University of Tuebingen.

## Author Contributions

**Conceptualization:** Matthias Weissinger, Jürgen Kupferschläger, Christian la Fougere, Gerd Grözinger.

**Data curation:** Matthias Weissinger, Jonas Vogel, Jürgen Kupferschläger, Ulrich Grosse, Christoph Artzner.

**Formal analysis:** Matthias Weissinger, Jonas Vogel, Christian la Fougere, Gerd Grözinger.

**Funding acquisition:** Konstantin Nikolaou, Gerd Grözinger.

**Investigation:** Matthias Weissinger, Jonas Vogel, Jürgen Kupferschläger, Helmut Dittmann, Gerd Grözinger.

**Methodology:** Jürgen Kupferschläger, Christian la Fougere, Gerd Grözinger.

**Project administration:** Gerd Grözinger.

**Supervision:** Jürgen Kupferschläger, Helmut Dittmann, Konstantin Nikolaou, Christian la Fougere, Gerd Grözinger.

**Validation:** Matthias Weissinger, Christian la Fougere, Gerd Grözinger.

**Visualization:** Matthias Weissinger, Jonas Vogel.

**Writing – original draft:** Matthias Weissinger, Salvador Guillermo Castaneda Vega, Gerd Grözinger.

**Writing – review & editing:** Jonas Vogel, Helmut Dittmann, Salvador Guillermo Castaneda Vega, Ulrich Grosse, Christoph Artzner, Konstantin Nikolaou, Christian la Fougere, Gerd Grözinger.

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
