## [Decision Letter · Decision Letter 0]

12 Oct 2020

PONE-D-20-23335

Correlation of C-arm CT acquired parenchymal blood volume (PBV) with 99mTc-macroaggregated albumin (MAA) SPECT/CT for radioembolization work-up

PLOS ONE

Dear Dr. la Fougère,

Thank you for submitting your manuscript to PLOS ONE. After careful consideration, we feel that it has merit but does not fully meet PLOS ONE’s publication criteria as it currently stands. Therefore, we invite you to submit a revised version of the manuscript that addresses the points raised during the review process.

A special consideration should be directed towards improving statistical methodology and writing an appropriate reasoning behind the renewed statistical analysis and accordingly improved figures. A clear statement on the validity of the re-calculated p-values should be also presented please.

We look forward to receiving your revised manuscript.

Kind regards,

Domokos Máthé

Academic Editor

PLOS ONE

Journal Requirements:

"This study was approved by our institutional review board (Decision No. 747/2014BO1)."

b. Once you have amended this statement in the Methods section of the manuscript, please add the same text to the “Ethics Statement” field of the submission form (via “Edit Submission”).

3. Please provide additional details regarding participant consent.

In the ethics statement in the Methods and online submission information, please ensure that you have specified what type you obtained (for instance, written or verbal, and if verbal, how it was documented and witnessed).

If your study included minors, state whether you obtained consent from parents or guardians.

If the need for consent was waived by the ethics committee, please include this information.

'The funders had no role in study design, data collection and analysis, decision to publish, or preparation of the manuscript.'

*Please include your amended statements within your cover letter; we will change the online submission form on your behalf.*

Reviewers' comments:

Reviewer's Responses to Questions

**Comments to the Author**

1. Is the manuscript technically sound, and do the data support the conclusions?

Reviewer #1: Yes

Reviewer #2: Partly

2. Has the statistical analysis been performed appropriately and rigorously? 

Reviewer #1: Yes

Reviewer #2: No

3. Have the authors made all data underlying the findings in their manuscript fully available?

Reviewer #1: Yes

Reviewer #2: No

4. Is the manuscript presented in an intelligible fashion and written in standard English?

Reviewer #1: Yes

Reviewer #2: Yes

5. Review Comments to the Author

Reviewer #1: The authors have set a clear goal for the paper: improve 99mTc/MAA SPECT-CT for predicting intrahepatic 90Y sphere distribution as a possible proxy for reducing the spread of embolization and leakage outside target tumors when treating HCC and CRC liver metastases. The stated improvement was assessing PBV maps as well from contrast C-Arm CT scans to reduce the high variance associated in MAA and 90Y correlation. The authors draw the line between conclusions and dicussion properly.

Suggestions for improvement:

line 156 "up to five respresentative...tumors...criteria for best visibility" and line 199 onwards: authors should indicate how they will clarify edge cases and poor visibility candidates and tumor size implications not included in the report could distort the statistics, i.e. false negatives, potentially visible as discrepancies in some components of patient survival. A potential beneficial way to do this is to connect with the text between lines 337 and 342 and 416+, where this is well rounded.

It would be good to improve readability by presenting a clear schematics/table on the size, microvasculation, contrast material perfusion, etc. issues that different methods have, and how to reconcile them statistically onto an overall prediction of required dose.

Is there any clinical benefit for the Bremsstrahlung validation that could improve MAA and PBV estimation further?

Reviewer #2: The manuscript does an excellent job when introducing the recent need for new approaches additionally to the 99mTc-MAA_SPECT/CT (Tc-Maa) to a better tumor perfusion prediction for preparing the transarterial radioembolization (TARE). Such an additional assistive technique they recommended the parenchymal blood volume imaging (PBV).

The main weakness of the article is that although the conclusions drawn seem valid, there are a number of major issues related to the statistical (and sampling) methods used. These are as follows.

In general

1. The most problematic part in the statistical evaluation is the ignoring of dependency between tumors among same patient and imaging. Consequently, the calculated p-values are not valid in the paper.

2. It is not clear what does „representative tumor” means. If tumors were not (almost) randomly selected, bias may arise.

3. Multiplicity correction was not mentioned.

4. Comparing correlation R-values especially in case of different sample sizes without mentioning an uncertainty (e.g. using confidence intervals) could be misleading.

5. I do not understand why to separate tumor size to categories <25 mm and > 25 mm instead of using their measured values.

Specific remarks

6. At line 215. It is not clear what parameters were compared with Wilcoxon signed-rank test: the distributions or the means/medians/etc. (with assuming symmetrical distribution).

7. At chapter Results, Tumor size. In my opinion here the question is rather an agreement than correlation.

8. At chapter „Optical lesion characterization of…” Because of the small sample size compared to the number of OLC categories I would prefer a more careful conclusion about OLC comparisons. (e.g. if we calculate the confidence interval for the mentioned proportions they will overlap in several cases)

9. At table 5 we could observe situations where the difference of „mean – standard deviation” resulting negative values. It would be better if it is explained why.

In addition to the statistical questions, I had the following questions and comments:

1. At chapter Image evaluation on page 168-169. what does „adapted if necessary” mean. Under what circumstances and how were they “adapted”.

2. At table 3 (about OLC values) we could observe group sizes, as long as the text shows percentages that is quite confusing.

3. About the „correlation of pre-therapeutic imaging with post-therapeutic 90Y-Brehmstrahlung-SPECT/CT”: instead of „correlation” I would prefer to say agreement or hit rate.

4. At the references: the citation number 21 is incomplete. There is no download or citation time given for citations number 29. and 30. (where the authors cite a website).

Overall, I find the article interesting and the results presented in it worthy for publishing, but the evaluation of the results is only possible with the appropriate statistics, so I definitely recommend improving them.

6. PLOS authors have the option to publish the peer review history of their article (what does this mean?). If published, this will include your full peer review and any attached files.

Reviewer #1: No

Reviewer #2: No

---

## [Author Response · Author response to Decision Letter 0]

26 Nov 2020

Reviewer(s)' Comments to Author:

Dear Reviewer #1,

Thank you for helpful comments and for taking the time to evaluation our manuscript. In order to address your various points we we adopted the following color code:

Black: Your comment

Green and marked with **: Our reply to your point.

Red and marked with “”: In-text changes following our reply to your point.

1. line 156 "up to five respresentative...tumors...criteria for best visibility" and line 199 onwards: authors should indicate how they will clarify edge cases and poor visibility candidates and tumor size implications not included in the report could distort the statistics, i.e. false negatives, potentially visible as discrepancies in some components of patient survival. A potential beneficial way to do this is to connect with the text between lines 337 and 342 and 416+, where this is well rounded.

**Thank you very much for this important comment. Here we try to better clarify the aim of this study, which was to compare the information of MAA-SPECT and C-arm CT based PBV maps in patients who are candidates for TARE procedure. Target lesions were selected based on contrast enhancement in diagnostic imaging (MR or CT), and most of our patients presented with less than 5 lesions. Lesions without contrast enhancement in MR or CT are supposed to be poor candidates for a TARE procedure. Therefore we have to agree, that we have a planned selection bias since we aimed to compare the methods (MAA and PBV) in those patients that were candidates for a TARE. However, intratumoral uptake pattern of MAA in SPECT or BBV maps from C-arm-CT were not used to decide to treat or not to treat one patient. Lesion size is reported in our analysis and of course we aimed to assess the most prominent lesions, since tumor mass in known to play a role for patient survival. 

False negative patients in CT and MR are therefore of course a confounding factor for overall survival, however at least in our institution; those patients would be excluded from a TARE procedure. 

As proposed, we have specified the way of defining the representative tumor and it to the text passage "image evaluation page 9, left 156".

Page 7, line 98-102: “Up to 5 representative tumors per patient were defined in the last contrast enhanced cross-sectional imaging and were evaluated in each patient on the basis of tumor size, best visibility and delineation (MRI n=15, CT n=18, 18F-FDG PET/CT n=1, interval 38 ± 21 days). In total, 102 tumors (54 CRC, 48 HCC) were defined accordingly and analyzed further using C-arm CT and SPET/CT. Comprehensive tumor characteristics are shown in suppl. table 1 “

2. It would be good to improve readability by presenting a clear schematics/table on the size, microvasculation, contrast material perfusion, etc. issues that different methods have, and how to reconcile them statistically onto an overall prediction of required dose.

**Thank you for the suggestion. We now included the requested table with the different parameters assessed within the study.

However, our data analysis didn’t enable us to make a fair correlation to post therapeutic dosimetry because of the limited image quality of Bremsstrahlen SPECT, that didn’t enable a accurate dosimetry quantification. Therefore only a descriptive correlation of post-therapeutic Bremsstrahlen-Scans with MAA or PBV maps was presented.

Our hypothesis was that absolute assessment of PBV and relative MAA-uptake might be of help for a more accurate prediction of required dose for TARE planning when the partition model is used. However as shown in our study, only few patients (HCC with capsular growing pattern) showed a good correlation between both methods. Furthermore, MAA was shown to be the better predictor for post therapeutic spheres deposition. Therefore, we decided not to implement PBV-data for dosimetry planning.

We have added a list of all tumor lesions as a supplemental table.** Line 102: “Comprehensive tumor characteristics are shown in suppl. table 1”

3. Is there any clinical benefit for the Bremsstrahlung validation that could improve MAA and PBV estimation further?

The limited information obtained by Bremsstrahlung scan in both quantification as well as resolution, limits its use for dosimetry. Therefore, we aim to include Y-90 PET that comes with higher resolution and might also enable a more accurate data quantification. However, even if PBV is a very powerful tool for treatment planning (e.g. vascular territories) its use for dosimetry seems to be limited because of the differences in tracer application (slow pulsatile) and contrast media application (bolus application) especially when SIR-Spheres are used. One the other hand this could be somewhat different when Terraspheres are used since the glass spheres are also applicated as a bolus.

Dear Reviewer #2,

Thank you for helpful comments and for your time evaluating the submitted manuscript. Herewith, we try addressing your points. For your convenience, we adopted the following color code:

Green and marked with **: Our reply to your point.

Red and marked with “”: In-text changes following our reply to your point.

In general:

1. The most problematic part in the statistical evaluation is the ignoring of dependency between tumors among same patient and imaging. Consequently, the calculated p-values are not valid in the paper.

**Thank you very much for this important comment. We have now revised and correctly recalculated our statistical analysis with our in-house statistician. 

By limiting the number of tumors to 5 per patient, we aimed to avoid that our effect might be driven by a few patients. We also agree that variabilities in the number of assessed tumors per patient may lead to some kind of clustering. 

In our cohort we found high intra-individual variabilities for both PBV and MAA. Therefore, we assume that our findings are not driven by the summation of intra-individual effects that would provide good correlations by mistake.

1. To verify the correlation of PBV and MAA-TBR, we performed the "repeated measures correlation," which accounts for non-independence among observations by adjusting for inter-individual variability using R. Using multiple measure corrections, a significant correlation could be confirmed between PBV and TBR for the capsular subtypes of HCC (r:0.45, p<0.05). It could also be confirmed that there is no correlation for CRCs. However, although the analysis was significant using Pearson correlation (in all HCCs diffuse and capsular) the new analysis was not in agreement. For a more detailed interpretation of the data, the 95% CI was added as recommended in point 4.

We have adjusted the methodological part according to the modified static tests:**

“Tumor size, TBR and MAA-Uptake was log-normal distributed, thus the statistical tests were performed on natural log-transformed data. For the comparison of PBV and 99mTc-MAA-Upake or TBR, linear regression with robust clustered standard error correction was applied using Stata 14 (StataCorp LLC, College Station , Texas, USA). Bland-Altman plots were calculated to analyze the agreement of tumor size measurements using SPSS (Version 27, IBM Corporation, Armonk, New York, USA). Exploratory data analysis was performed using SPSS and JMP® (Version 13.1, SAS Institut Corporation, Heidelberg, Germany). 

Because of the data structure, for the determination of the overall within-individual relationship among paired measures (PBV and TBR) within one patient, repeated measures correlations were performed with multiple measurement correction using rmcorr-package for R (V0.4.1.by J.Bakdash and L. Marusich) 

Rules of thumb was applied to interpret correlation coefficient rho: 0.20-0.39: weak; 0.40-0.59: moderate; 0.60-0.79: strong; 0.80-1.0: very strong.

Significance level of P values was 0.05. All values are expressed as mean values ± standard deviation with 95% confidence intervals are given in brackets.” 

**Distribution of log normal transformed data are presented as supplemental figure 1 a-d.**

**The cluster corrected correlations were recalculated using R. The figure 5 was replaced accordingly and the new results were inserted in the results section.

Page 19, line 355-359: **“PBV values showed a weak but significant correlation to 99mTc-MAA-TBR in HCC with capsular growth pattern (r=0.45, p<0.05), but no correlation in CRC-metastases independent of the growth pattern (r=0.1, p=0.54) as presented in Fig 5. HCC with diffuse growth pattern presented with negative correlation of tumors within the same patient but weak overall correlation (r cluster corrected: -0.18, r overall: 0.21)”

**Discussion was modified according the revised results: Page 22, linke 387-389:**

“Our study revealed significant differences between the distribution patterns of the contrast agent Iopromide used for PBV and the 99mTc-MAA particles for SPECT/CT.

This was observed not only in tumors of various patients, but also with individual tumors of the same patient. Only HCCs with capsular growth patterns showed a weak correlation between the measurement methods.”

**The results section in the abastact was also adapted to the revised results: **

“102 HCC or CRC metastases were analyzed. HCC presented with significantly higher MAA-TBR (7.6 vs. 3.9, p<0.05) compared to CRC. Tumors showed strong intra- and inter-individual dissimilarities between TBR and PBV with a weak correlations for capsular HCCs (r= 0.45, p<0.05) and no correlation for CRC.”

2. **Furthermore, we checked the statistical evaluation regarding statistical differences between the measurement methods and the tumors (table 5) on the effect of multiple measurements. For this purpose we first checked the distribution of the data again. Using log-transformation we obtained a log-normal distribution. As a consequence, we were able to apply a linear regression with robust clustered errors correction instead of the Wilcoxon test. For the log-transformed data the null hypothesis could be rejected for the linear regression of the MAA-TBR of HCC and CRC. However this analysis led to a non-significant result of the ln(PBV) data (p=0.08) in contrast to the non-log transformed data (p=0.03). The impact of tumor growth (diffuse or capsular) on the liver Tc-MAA background uptake could be confirmed.

Table 5 was reworked and the new results were inserted in the results section, however the main results did not changed. 

Page 18, line 345-350:** “The capsular HCCs had two outliers with 846 and 719kBq/cm³. Smaller tumors (<25mm) presented only a non-significant trend towards higher PBV-values (p=0.12). A diffuse growth pattern of the tumor was associated with a significantly higher 99mTc-MAA background (normal liver tissue) when compared to a capsular growth pattern, resulting in lower 99mTc-MAA-TBR values.”

3. We have provided the log-transformed data in an additional table as supplementary data. For easier readability we would like to keep the non log-transformed data in the table in the main manuscript, but provided the 95% CI instead of the mean value as the data are skewed. 

Log-tranformed data are presented in supplemental table 2. Page 18, line 341-342:

“Significance was tested on the log-transformed transformed data, which are shown in supplementary table 2”

2. It is not clear what does „representative tumor” means. If tumors were not (almost) randomly selected, bias may arise.

Thank you very much for pointing out this detail in our methodology. Reviewer 1 made a similar observation. We have bound are responses to both reviewers in the following statements:

**Thank you very much for this important comment. Here we try to better clarify the aim of this study, which was to compare the information of MAA-SPECT and C-arm CT based PBV maps in patients who are candidates for TARE procedure. Target lesions were selected based on contrast enhancement in diagnostic imaging (MR or CT), and most of our patients presented with less than 5 lesions. Lesions without contrast enhancement in MR or CT are supposed to be poor candidates for a TARE procedure. Therefore we have to agree, that we have a planned selection bias since we aimed to compare the methods (MAA and PBV) in those patients that were candidates for a TARE. However, intratumoral uptake pattern of MAA in SPECT or BBV maps from C-arm-CT were not used to decide to treat or not to treat one patient. Lesion size is reported in our analysis and of course we aimed to assess the most prominent lesions, since tumor mass in known to play a role for patient survival. 

False negative patients in CT and MR are therefore of course a confounding factor for overall survival, however at least in our institution; those patients would be excluded from a TARE procedure. 

As proposed, we have specified the way of defining the representative tumor and it to the text passage "image evaluation page 9, left 156".

Page 7, line 98-102: “Up to 5 representative tumors per patient were defined in the last contrast enhanced cross-sectional imaging and were evaluated in each patient on the basis of tumor size, best visibility and delineation (MRI n=15, CT n=18, 18F-FDG PET/CT n=1, interval 38 ± 21 days). In total, 102 tumors (54 CRC, 48 HCC) were defined accordingly and analyzed further using C-arm CT and SPET/CT. “

3. Multiplicity correction was not mentioned.

**Instead of making multiple comparisons, we have opted, in the revision, to use linear regression to clarify the relationship between Uptake, PBV and size. The linear regression was performed on log-normalized data, and standard errors as well as p-values were corrected for the non-independence of observations (cluster corrected, using STATA13) in Table 5 as shown in the manuscript. **

4. Comparing correlation R-values especially in case of different sample sizes without mentioning an uncertainty (e.g. using confidence intervals) could be misleading.

**As mentioned in point 1, we have re-evaluated the correlations with an additional multiple measurement correction. As recommended, we have added the 95% CI for both normal and log-transformed data. As expected, the 95% confidence interval was relatively broad in the relatively small group of HCC with diffuse growth pattern.

Regarding the changes in the manuscript we would like to refer to point 1.**

5. I do not understand why to separate tumor size to categories <25 mm and > 25 mm instead of using their measured values.

**For the comparison of PBV and MAA-TBR with bremsstrahlung SPECT as the defined gold standard, the classification into tumors smaller and larger than 25mm was important. Due to the limited resolution of bremsstrahlung SPECT, a lesion size of at least 25mm is essential for an adequate evaluation. Therefore, our primary aim was to test whether there are significant differences between larger and smaller lesions and, if so, whether these results may be extrapolated from large tumors to small tumors.**

Specific remarks

6. At line 215. It is not clear what parameters were compared with Wilcoxon signed-rank test: the distributions or the means/medians/etc. (with assuming symmetrical distribution).

**After the data had been successfully checked for a log-normal distribution, the Wilcoxen test was discarded and parametric tests were applied instead.**

7. At chapter Results, Tumor size. In my opinion here the question is rather an agreement than correlation.

**Thank you for this remark. We agree and corrected according to the reviewers suggestion. The agreement of the size measurements was checked with a Bland Altman plot. Therefore the statics in Material and Methods was adapted accordingly. 

Figure 2 was replaced by Bland-Altman plots, which were described in detail in the results section as follows: Page 13, line 240-250.** “In particular, CRC are displayed slightly too small in PBV maps compared to CT or MRI with an average discrepancy of 1.5mm as presented in Figure 2a. However, differences of measurements are within relative narrow limits (-7.0 to 10.0mm) and without a trend as mean of both measurements increase.

Measuring tumor size of HCC seems to have a perfect agreement with a mean difference of 0.1mm between both methods (figure 2b). However, due to single outliers, the range for the 1.95 standard deviation limit was quite large (from -19.8 to 20.1mm), and was even exceeded by some outliers in both directions. However, most of the differences are within a narrow limit and without a trend increasing size related measurement differences or variabilities. Although our data are not distributed normally, the differences seem to be.”

8. At chapter „Optical lesion characterization of…” Because of the small sample size compared to the number of OLC categories I would prefer a more careful conclusion about OLC comparisons. (e.g. if we calculate the confidence interval for the mentioned proportions they will overlap in several cases)

**The fine scaling of the OLCs defined in the methodology resulted in relatively small subgroups. By re-grouping the subgroups with similar characteristics (4+5 or 0+1) we have tried to counteract this effect. In Table 3 we also present a detailed overview of the case numbers. 

As requested by the reviewer we have put the evaluation in the discussion into perspective according to the limited statistical power with small sample size.** Page 24: line 446 However, the statistical power was limited due to the small sample size.

9. At table 5 we could observe situations where the difference of „mean – standard deviation” resulting negative values. It would be better if it is explained why.

**The measurement of the MAA uptake of the capsular HCCs shows 2 outliers with 3 and 4 times of the mean uptake. This uptake is relativized by calculating a tumor to background ratio. These measured data were checked again to rule out a measurement error. As our data are log-normal distributed, we corrected the standard deviation into the now presented 95% CI.

We described the outliners in the results section page 18, line 345**: “The capsular HCCs had two outliers with 846 and 719kBq/cm³.”

**To visualize the outlier an additional scatterplot is presented in the supplementary data section.**

In addition to the statistical questions, I had the following questions and comments:

1. At chapter Image evaluation on page 168-169. what does „adapted if necessary” mean. Under what circumstances and how were they “adapted”.

**In the case of multiple neighboring or unclearly edged tumors in the PBV map or MAA-SPECT, the images were coregistrated on the previous images (MRI/CT) in order to correlate the tumors precisely.

he text was adjusted accordingly: Page 10, line 166-168** : “Moreover, in order to ensure a high conformity to the tumor-morphology, every VOI was coregistered to the corresponding pre-interventional CT or MRI and carefully adapted in size if necessary.”

2. At table 3 (about OLC values) we could observe group sizes, as long as the text shows percentages that is quite confusing.

**The percentage has been replaced by the absolute number in the text to avoid misunderstandings. Furthermore, as criticized in point 8, the result can now be interpreted in relation to the case numbers.

Page 15, line 285-291**: “OLC evaluation of PBV and 99mTc-MAA values were consistent (exact: 52/102) same category of intensity or homogeneity: 37/102), independent of tumor size, growth pattern or tumor entity. However, the amount of homogeneous intense uptake within the metastases (OLC 5) was less pronounced for the PBV-maps (30/102) when compared to 99mTc-MAA-scans (43/102).”

3. About the „correlation of pre-therapeutic imaging with post-therapeutic 90Y-Brehmstrahlung-SPECT/CT”: instead of „correlation” I would prefer to say agreement or hit rate.

**In figure 4, "correlation" has been replaced by "agreement" in the manuscript as suggested.

Results section page 13, line 235:** “PBV-maps enabled an accurate assessment of the tumor size, with a good agreement to the pre-therapeutic CT and MRT-scans (C-arm CT 36.7 ± 27.2mm vs. CT/MRT 35.9 ± 24.4mm). “

Results section page 13, line 244: “Measuring tumor size of HCC seems to have a perfect agreement with a mean difference of 0.1mm between both methods (figure 2b).”

Figure legend 4: “Agreement of pre-therapeutic imaging with post-therapeutic 90Y-Bremsstrahlung-SPECT/CT. Good visual correlation agreement of 99mTc-MAA uptake and PBV regarding the small lesion in segment VIII (y), but not for the bigger lesion in segment VII (x), especially the ventral part of the lesion (x1).”

4. At the references: the citation number 21 is incomplete. There is no download or citation time given for citations number 29. and 30. (where the authors cite a website).

We appreciate your attention. 

Reference 21 (conference abstract) was replaced by the original paper published in AJNR.

Furthermore we have manually added the citation time to the literature management software output.

---

## [Editor Report · Decision Letter 1]

7 Dec 2020

Correlation of C-arm CT acquired parenchymal blood volume (PBV) with 99mTc-macroaggregated albumin (MAA) SPECT/CT for radioembolization work-up

PONE-D-20-23335R1

Dear Dr. la Fougere,

We’re pleased to inform you that your manuscript has been judged scientifically suitable for publication and will be formally accepted for publication once it meets all outstanding technical requirements.

Kind regards,

Domokos Máthé

Academic Editor

PLOS ONE

Additional Editor Comments (optional):

With the reviewer responses carefully built into the text, the manuscript confers important and practically, clinically usable information for personalized tumor therapy. I think it is a welcome addition to PLOS ONE.

It would be interesting if the same well-built and meticulous team performed similar studies using 166Ho or 177Lu, i.e. isotopes with good SPECT resolvability. Also, I would welcome one small sentence in the conclusion part pointing out that fully quantitative SPECT is nowadays a reality and could also enhance clinical outcomes if more studies like this one, now proposed to be accepted, appear.

Reviewers' comments:

As the Academic Editor, I took the task to overview and consolidate the reviewer opinions and the re-written manuscript. I found that the Authors satisfactorily corrected and amended the manuscript, which now indeed offers a good precision medicine outlook with "old school" means and the smart use thereof. It is proposed to be published.

---

## [Editor Report · Acceptance letter]

18 Dec 2020

PONE-D-20-23335R1 

Correlation of C-arm CT acquired parenchymal blood volume (PBV) with ^99m^Tc-macroaggregated albumin (MAA) SPECT/CT for radioembolization work-up 

Dear Dr. la Fougere:

I'm pleased to inform you that your manuscript has been deemed suitable for publication in PLOS ONE. Congratulations! Your manuscript is now with our production department. 

Kind regards, 

on behalf of

Dr. Domokos Máthé 

Academic Editor

PLOS ONE